# The Composition of Fatty Acids in Bee Pollen, Royal Jelly, Buckthorn Oil and Their Mixtures with Pollen Preserved for Storage

**DOI:** 10.3390/foods12173164

**Published:** 2023-08-23

**Authors:** Violeta Čeksterytė, Saulius Bliznikas, Kristina Jaškūnė

**Affiliations:** 1Institute of Agriculture, Lithuanian Research Centre for Agriculture and Forestry, Instituto al. 1, 58344 Kėdainiai, Lithuania; kristina.jaskune@lammc.lt; 2Institute of Animal Science, Lithuanian University of Health Sciences, R. Zebenkos 12, 82317 Baisogala, Lithuania; saulius.bliznikas@lsmu.lt

**Keywords:** pollen, honey, sea buckthorn oil, royal jelly

## Abstract

Honey produced by *A. mellifera* contains minor components present in the nectar collected from plants. Various studies of honey components and all other bee products can be informative in assessing their quality. The aim of the present study was to determine the content and composition of fatty acids (FAs) in sea buckthorn oil (SBO), royal jelly (RJ) and bee pollen (BP) and the changes in FAs content in these products during storage. The diversity of FAs and the effect of storage time on FAs content was also evaluated for the prepared-for-preservation mixtures, which included the following samples: pollen mixed with honey at a ratio of 1:2 (*w*/*w*); sample BPH, a well; BPH + 1% (*w*/*w*) SBO; and BPH + 1% (*w*/*w*) SBO + 2% (*w*/*w*) RJ. Fresh bee-collected pollen and RJ were stored at −20 °C, whereas the conserved samples were stored at +4 °C in hermetically sealed jars. The data revealed that RJ demonstrated the highest diversity of fatty acids compared to BP and BP prepared for storage with honey along with SBO and RJ. Palmitic and stearic acids were found in the highest amounts out of the eight saturated fatty acids identified in the studied SBO and RJ. The amount of these fatty acids in RJ compared to SBO was 1.27 and 6.14 times higher, respectively. In total, twenty-two unsaturated fatty acids (USFA) were identified in RJ and fourteen were found in SBO. The SBO used in this study was found to be high in linoleic acid, resulting in an increased n-6 fatty acids ratio in the prepared samples. Essential fatty acids eicosapentaenoic (EPA) and docosahexaenoic (DHA) were found in RJ, as well as in BP and BP mixed with honey. These FAs were not identified in the samples prepared with SBO even in the sample supplemented with RJ. The highest decrease in docosadienoic fatty acid was found in the BPH sample compared to BP, while arachidonic acid mostly decreased in BPH + 1% SBO compared to the BPH + 1% (*w*/*w*) SBO + 2% (*w*/*w*) RJ samples stored at +4 °C. Bee-collected pollen had the greatest influence on the number of FAs in its mixture with honey.

## 1. Introduction

The nutritional value of honey is determined by sugars, proteins, enzymes, amino acids, phenols, vitamins, aroma compounds and other substances present in honey [1]. The composition of the variety of compounds in honey determines its biological properties such as its antimicrobial, antioxidant, antiviral, antifungal and anti-inflammatory properties, among others [2,3]. It has been revealed that honey inhibits the activity of about 60 bacteria, which are aerobic, anaerobic, Gram-positive and Gram-negative [4]. Honey is used to preserve food due to its antibacterial properties [5]. The main compound responsible for honey’s antibacterial effect is believed to be hydrogen peroxide. Hydrogen peroxide in honey is mainly formed during the oxidation of glucose, which is catalyzed by the bee enzyme, glucose oxidase (GOD) [6]. GOD was detected in honey of different origins, and the amount of hydrogen peroxide produced by GOD and antimicrobial activity varied among honey samples [7]. Among other components attributable to the non-peroxide antimicrobial activity of honey are bee-secreted proteins, such as major royal jelly (MRJ) proteins (1–5) [8], defensin-1 [9,10] and hymenoptaecin [10]. The antibacterial properties of honey are employed in the food industry for prolonging the shelf life of the production; thus, they are often added in juice or dairy products, such as yogurt for activating bifidobacteria [11,12]. According to our recent studies, GOD was found in fresh bee pollen, dried bee bread, pollen–honey and bee bread mixtures with honey, along with vegetable oils that have been used as supplements in bee products. The activity of this enzyme was higher in the studied bee bread compared to pollen, as well as in all bee bread mixtures with honey and added vegetable oils. The data revealed that fresh pollen and all conserved mixtures with fresh pollen can be stored in a refrigerator at +4 °C room temperatures for one year [13]. Other substances such as honey sugars and organic acids—i.e., gluconic, acetic, lactic, formic and propionic, among which gluconic acid is predominant—have an influence on honey preservation and the physicochemical and sensory properties of honey [3,14].

The content of lipids in fresh and lyophilized royal jelly (RJ) amounts to 4–8 and 15–30%, respectively [15]. RJ is a product secreted by bees from their hypopharyngeal glands (HPGs) and fed to the bee queens and larvae [16]. Researchers point out that fatty acids (FAs) present in RJ are mostly composed of medium chain fatty acids, containing 8–12 carbon atoms [17]. Among FAs, *trans*-10-hydroxy-2-decenoic acid (10-HDA) is the main monounsaturated fatty acid (MUFA), accounting for more than 70% of the total fatty acids in RJ [18,19]. According to the requirements of the International Organization for Standardization, 10-HDA content for RJ should be more than 1.4% of the total RJ composition [20]. The 10-HDA is studied mostly via all components of a lipid fraction in RJ. For a long time, these FAs were considered to be a key factor in the antibacterial activity of RJ; however, recently, this role for the above FAs has been questioned [21,22], as more detailed RJ lipid profiles of different botanical origins have been determined [15]. The authors identified 11 FAs in RJ samples which were collected from *Apis mellifera* colonies fed by different plant pollen. Among them, decanedioic acid and 2-decenoic acid were the most abundant FAs in all RJ samples. According to the data of this study, it is determined that phospholipids and MUFAs have the greatest influence in differentiating RJs samples of different botanical origins. The findings showed that FAs content and the quality of RJ depends on the type of pollen on which bee colonies are fed [23]. Pollen collected by bees is necessary for honeybee colonies to produce RJ [15]. Pollen lipids make up to 13%, and their content and variety depend on the plant source [24]. Studies show that the contents of 18-carbon-saturated and unsaturated FAs in RJ change when feeding bees with single bee pollen derived from *Acer mono* Maxim. and *Phellodendron amurense* Rupr. The data indicate that the best quality RJ was collected after 18 or 24 days of bee feeding with selected pollen. A diet with single pollen for bees may destroy their nutritional balance; therefore, pollen mixture is recommended as a better feed for bees [23].

FAs in monofloral bee bread (BB) and bee pollen (BP) loads collected from *Trifolium pratense* L. were identified and quantified via gas chromatography, indicating that the content of two FAs possessing n-3 and n-6 structures was found to be the highest [25], where α-linolenic (C18:3n-3) in BB and BP varied in the range of 38.3–45.1% and 20.4–35.5%, respectively. The value of linoleic acid (C18:2-n-6) in BB and BP was 4.74 ± 0.13% and 7.57 ± 1.32%, respectively. Docosahexaenoic acid (DHA) was presented both in monofloral BB and BP from *T*. *pratense*. Bee products containing health-benefiting n-3 fatty acids are exceptional because these FAs cannot be synthesized in the human body and must be obtained through food intake. The enzymes enlongase and desaturase convert alpha-linolenic acid into EPA and DHA, though they produce a relatively small amount of these FAs [26]. Most studies have confirmed that intake of food with higher n-3 fatty acid content is associated with better body fitness [27,28,29].

Bees collect pollen from different plants, so the content of FAs in this product varies [30,31]. Among saturated FAs found in Romanian BP, the predominant ones are palmitic (C16:0) and stearic (C18:0) acids [31]. The same FAs dominated in the BP from 11 different floral sources from Taiwan [32]. Among the saturated FAs found in bee pollen samples collected in the Tuscany region of Italy, three dominated, including palmitic acid (C16:0), stearic acid (C18:0) and margaric acid (C17:0) [33]. Fatty acids composition was determined in Brazilian honey BP. The unsaturated fatty acid (USFA) content in Brazilian pollen varied from 18.6% to 55.9% of the total FA content. Researchers believe that pollen is a good source of USFA [34], suggesting that the ratio between USFA and SFA is of great importance. Pollen is considered to have high nutritional value when the USFA/SFA ratio is greater than 1. An USFA/SFA value less than 1 indicates reduction of USFA due to storage and dehydration processes [35]. The USFA/SFA ratio ranged from 2.2 to 6.7 in the pollen collected in India and from 1.91 to 5.86 in the pollen collected in Portugal [36,37]. In addition, the ratio of FAs n-3/n-6 is also an important criterion for evaluating the health-beneficial properties of bee pollen. The value of n-3/n-6 ratio ranged from 0.06 to 3.09 in Indian pollen and from 0.17 to 0.52 in Portuguese pollen. Based on the review of the conducted research, the studies show that the diversity and differences of FAs present in bee-collected pollen can be attributed to their origin and the influence of geographical conditions, pollen harvest season and pollen quality, as well as the methodologies used for their identification [35,38,39].

Sea buckthorn oil is extracted from the berries, seeds and peel of the sea buckthorn plant (*Hippophae rhamnoides*) [40]. Usually, the oil obtained from the pulp/peel fraction is blended due to their separation difficulties [41]. The berries’ pulp and the seed yielded a nutritious oil of different FAs composition [42]. The main FAs present in seed oil are linoleic (C18:2 n-6), α-linolenic (C18:3 n-3), oleic (C18:1 n-9) and palmitoleic (C16:1 n-7) acids, constituting approximately 40%, 30%, 16% and 0.5% of the total FAs in seed oil, respectively [42,43]. Palmitoleic fatty acid is more abundant in peel and pulp oil, accounting for 24% to 36% of the total FAs, respectively [44]. Palmitoleic acid is known to be valuable for skin care as it supports cellular tissue and wound healing [45]. In pulp oil, the major saturated fatty acids are palmitic acid (C16:0) and stearic acid (C18:0), [42]. Palmitic and stearic acids accounted for 11.23–19.12% and 2.15–2.88%, respectively, in the oil of the Indian sea buckthorn seed species [46]. The presented data on sea buckthorn oil show that the difference between seed and pulp oil lies in the relatively high amount of C16 fatty acids in the pulp oil and the relatively high content of C18 fatty acids in the seed oil. The FAs composition of sea buckthorn oil can be used in plant breeding to assess the nutritional value of their production [46].

The aim of the study was to (1) evaluate the composition of FAs in sea buckthorn oil, royal jelly and bee-collected pollen, as well as in the mixtures of honey with pollen and with pollen and royal jelly; (2) evaluate the changes in the FAs content and composition in the studied products stored for two years under specified conditions; (3) determine the influence of bee pollen FAs content on the FAs content in the pollen mixture with honey after a two-year period of storage.

## 2. Materials and Methods

### 2.1. Preparation of Bee-Collected Pollen and Bee Pollen Compositions with Honey and Additives

Bee pollen (BP) was collected in the apiary of the Institute of Agriculture, Lithuanian Research Centre for Agriculture and Forestry. Pollen loads were collected from early spring to mid-July at the divisions of apiaries situated at different sites of Kėdainiai district, Lithuania. Part of the bee-collected pollen was used for conservation; pollen was mixed with honey at a ratio of 1:2 (*w*/*w*) (sample BPH). Sea buckthorn oil (SBO) (1% *w*/*w*) was added to one part of BPH (sample BPH + 1% (*w*/*w*) SBO). To another part of BPH, 1% (*w*/*w*) sea buckthorn oil (SBO) and 2% (*w*/*w*) royal jelly (RJ) were added (sample BPH + 1% (*w*/*w*) SBO + 2% (*w*/*w*) RJ). All samples used for tests and the concentrations of the additives SBO and RJ are presented in Table 1, Table 2, Table 3, Table 4 and Table 5. Fresh BP and RJ were stored in the freezer at −20 °C, while the conserved samples with honey and additives were stored at +4 °C in hermetically sealed jars. Sea buckthorn oil was purchased at a pharmacy (sea buckthorn oil producer SALVENA S.C., Krakow, Poland) and stored at +4 °C.

### 2.2. Microscopic Examination of Pollen Samples

Pollen sample preparation for botanical composition analysis was performed using the method of Louvelaux et al. [47]. Pollen grains were identified with a Nikon Eclipse E600 microscope, model C-LP (Nikon Corporation, Tokyo, Japan). Pollen photos were taken under a Nikon Eclipse E600 microscope and compared with the known plant pollen photos presented in the pollen catalogue [48]. Pollen botanical composition was assessed by calculating the frequency of pollen in samples and expressed as a percentage of total pollen sum.

### 2.3. Chemicals

Hydrochloric acid, potassium chloride, sodium sulfate anhydrous, methanol, hexane, chloroform and sodium methoxide solution were purchased from Sigma-Aldrich, (Sigma-Aldrich, Burlington, MA, USA) [49]. Supelco 37 comp. FAME mix was purchased from Supelco (Supelco Analytical, Bellefonte, PA, USA).

### 2.4. Determination of Fatty Acids

The extraction of lipids for FAs analysis was performed with chloroform/methanol (2:1 *v*/*v*) as described in ref. [50]. Fatty acid methyl esters (FAMEs) of the total lipids were prepared according to the procedure described by Christopherson and Glass [51]. The FAMEs were analyzed using a gas chromatograph Shimadzu GC—2010 Plus (Shimadzu Corp., Kyoto, Japan) fitted with a flame ionization detector. The separation of methyl esters of fatty acids was performed on a capillary column Rt-2560 (100 m; 0,25 mm ID; 0,25 μm df) (Restek, Bellefonte, PA, USA) by temperature programming from 160 °C to 230 °C. The temperature of the injector was set at +240 °C, and the detector was set at +260 °C. The rate of flow of carrier gas (nitrogen) was 0,79 mL/min. The total GC analysis time was 60 min. The injection volume was 1.0 μL. The FAMEs were identified by comparing their retention times with those of the authentic standard mixtures: Supelco 37 comp. FAME mix. The relative content of each FA in the sample was expressed as a relative percentage of the sum of the FAs [50,51].

### 2.5. Statistical Anlysis

The analyses of FAs were carried out in three replicates. Estimated values are expressed as means ± standard error (SE). The data of measured indicators during product storage were evaluated using statistical Analysis of Variance (ANOVA). The significance levels are expressed at *p* ≤ 0.05, a confidence level of 95%. The correlation coefficients were calculated with MS Excel software using a paired regression statistic type of analysis.

## 3. Results

### 3.1. Botanical Composition of the Pollen Loads

Melissopalynological analysis was used to determine the percentage of pollen in the samples produced for the study. The botanical origin of plant pollen of each sample was compared with the data related to the standardized criteria used to assess the botanical origin of honey. The pollen of rapeseed (*Brassica napus*) dominated in samples of BPH, BPH + 1% (*w*/*w*) SBO and BPH + 1% (*w*/*w*) SBO +2% (*w*/*w*) RJ, constituting 46.3% 41.2% and 49.0%, respectively (Figure 1).

According to the method of melissopalynology, the pollen group is classified as important minor pollen, constituting 3–15% [47]. Among the important minor pollen were caraway (*Carum carvi*), white clover (*Trifolium repens*) and red clover (*Trifolium* pratense), which were present in the range of 6.2% to 12.0%. The contents apple-tree pollen (*Malus domestica*) in samples BPH and BPH + 1% (*w*/*w*) SBO + 2% (*w*/*w*) RJ were 3.5% and 3.0%, respectively. The contents of maple pollen (*Acer campestre*) in samples BPH, BPH + 1% (*w*/*w*) SBO and BPH + 1% (*w*/*w*) SBO + 2% (*w*/*w*) RJ samples were 2.9%, 3.2% and 3.6%, respectively. Pollen of buckwheat (*Fagopyrum esculentum*) constituted 3.2% in BPH + 1% (*w*/*w*) SBO and 3.1% in BPH + 1% (*w*/*w*) SBO + 2% (*w*/*w*) RJ. Willow (*Sinapis alba*) pollen was found to be 3.7% and 3.6% in the same respective samples. Pollen of coltsfoot (*Tussilago farfara*) was present at only 5.5% in BPH + 1% (*w*/*w*) SBO and less than 3.0% in other samples along with other identified pollens.

### 3.2. Fatty Acid Content of Sea Buckthorn Oil (SBO) and Royal Jell (RJ) Used for Pollen–Honey Mixture Additives

A nutritious oil of various FAs compositions is obtained from the berries’ pulp and the seeds of the sea buckthorn [42]. The composition and amount of saturated fatty acids (SFA) in SBO and RJ are shown in Table 1. Palmitic and stearic FAs were found at the highest amount among the eight SFAs identified in the studied SBO and RJ. The amount of these fatty acids in RJ compared to SBO was 1.27 and 6.14 times higher, respectively. Caprylic (octanoic) acid (C8:0) presented in RJ was not found in other tested bee products or SBO. The odd-chain fatty acid, pentadecanoic C15:0, was identified in both SBO and RJ, while margaric acid (C17:0) was not found in RJ but was present in a small amount (0.04%) in SBO. The content of fatty acid C15:0 was eight times higher in RJ than in SBO.

#### 3.2.1. Characteristics of Unsaturated Fatty Acids in SBO and RJ

In total, twenty-two USFAs were identified in RJ, while fourteen were found in SBO (Table 2). The conducted study revealed the diversity of USFAs and their different amounts in the studied samples.

The SBO had a significantly higher (*p* ≤ 0.05) content of USFAs 16:1n-7, C17:1n-9, C18:1n-7 and C18:2n-6*cis* compared to RJ. The RJ had significantly higher (*p* ≤ 0.05) levels of C18:1n9-*trans*, C18:1n-9*cis*, C18:3n-3, C20:1n-9, C20:5n-3 and C20:4n-6. The largest USFA quantitative difference between SBO and RJ was found for C18:3n-3 (ALA) and C18:2n-6c acids. The content of ALA with the n-3 structure (C18:3n-3) accounted for 4.35% of the total amount of RJ fatty acids, and it was 10.0 times higher than in those of the SBO. The highest level of n-6 linoleic fatty acid (C18:2n-6cis) was determined in SBO and constituted 57.16%. The latter data show the opposite distribution of n-3 and n-6 acids in the SBO and RJ as well as in the seeds of plants.

Monounsaturated fatty acids, such as C16:1n-7*trans*, C16:1n-9 and polyunsaturated fatty acids, with an n-3 structure including C20:3n-3, C22:5n-3, C22:6n-3 and n-6 fatty acids, such as C18:3n-6, C20:3n-6, C22:2n-6 and C22:4n-6, were identified in RJ but not in SBO. Valuable n-3 USFAs eicosapentaenoic (EPA) and docosahexaenoic (DHA) were present in RJ in small amounts of 0.27% and 0.69%, respectively. According to our results, EPA was also detected in SBO, albeit at the lower level of 0.18% compared to RJ.

#### 3.2.2. The Ratio of Saturated and Unsaturated Fatty Acids in SBO and RJ

The obtained research data on SBO show that the average content of n-3 fatty acids was low, accounting for only 0.61%, while the content of n-6 fatty acids was higher, making up 57.67% (Table 3).

The ratio of the latter fatty acids, namely n-3/n-6 and n-7/n-9, was 0.01 and 0.83, respectively, while the number of USFAs from the family of n-7 and n-9 was 2 and 6, respectively. Among the n-7 and n-9 FAs present in SBO, the highest level was found for palmitoleic acid (C16:1n-7) and oleic acid (C18:1n-9cis), namely, 9.11 ± 0.12% and 13.03 ± 0.15%, respectively. The average content of USFAs compared to SFAs was 5.10 times higher.

The total sum of n-3 fatty acids was 6.94% in RJ. The content of n-3 fatty acids varied in the range from 0.26% to 4.48%. The ratio of n-3/n-6 was 0.91 and indicated beneficial composition of FAs in RJ.

The average amount of SFAs in RJ was close to that of USFAs, resulting in an SFA/USFA ratio of 1.11. The numbers of n-7 and n-9 FAs were determined to be 2 and 6, respectively, and the average content of n-9 FA was 12.1 times higher compared to that of n-7 FA.

#### 3.2.3. Impact of Storage Time on the Changes of Fatty Acid Content in Royal Jelly

The data showed that RJ storage at −20 °C for 18 months had negatively affected the amount of thirteen FAs. The FAs content had changed significantly (*p* ≤ 0.05) but least for C18:2n-6*cis* and C20:3n-3. After 18 months, the residual contents for the above FAs were 82.6% and 63.4%, respectively (Figure 2).

While the FAs contents for C17:1n-9 and C20:5n-3 were close to half compared to their initial amounts of 48.9% and 48.1%, respectively, six FAs showed a greater decrease in content ranging from 39.3% to 17.8% compared to their initial amounts. The residual amount of FAs was less than 10.0%, determined for C22:5n-6, C16:1n-7 and C20:3n-3, respectively, as 8.89%, 5.2 6% and 3.14%. The detected amount of SFA did not decrease for five of the the eight and for twelve of the twenty-two USFAs (*p* ≤ 0.05). The current study shows that storage of RJ at −20 °C for 1.5 years results in different changes in the amount of each FA.

### 3.3. The Influence of Honey, Sea Buckthorn Oil and Royal Jelly on the Fatty Acids Content during the Storage of Preserved Bee Products

#### 3.3.1. Comparative Content of Pollen Fatty Acids and Pollen Preserved with Honey during Storage for Half a Year

Honey added to the pollen at a of ratio 1:2 did not change the FAs composition of the existing mixture (Table 4). FA amounts in the pollen–honey sample (BPH) compared to pure pollen (BP) were significantly different for 12 of the 24 identified FAs after half a year’s storage in a refrigerator at +4 °C. However, no significant differences were found in the FAs content between the samples BP and BPH at *p* ≤ 0.05 after two years of storage.

A lower content of FAs was determined for C12:0, C18:2n-6*trans*, C18:2n-6*cis*, C18:3n-3, C20:0, C20:3n-3 and C22:2n-6, while a higher content was determined for C18:0, C18:1n-9*cis*, C18:1n-7, C20:4n-6 and C22:6n-3 in the BPH sample compared to BP. Although the changes in the quantities of FAs were significant, the reduction of C18:2n-6*cis*, C18:3n-3 and C20:0 acids were slight in comparison to BP, accounting for 0.4%, 2.77% and 0.14%, respectively, while the remaining amount varied from 80.1% to 96.12%. The content of the FAs C20:3n-3 and C22:2n-6 decreased to 56.8% and 48.0%, respectively. The content of C18:3n-3 in samples BP and BPH was almost the same, accounting for 43.0% and 40.24%, respectively, and the differences between the amounts were insignificant. The content of DHA (C22:6n-3) was higher in BPH compared to BP, accounting for 0.19% and 0.93%, respectively. The data show that honey slightly alters the amounts of some FAs in the BPH mixture compared to BP and increases the content of the essential fatty acid DHA in the produced BPH mixture.

The content of FAs in the samples BPH + 1% SBO and BPH + 1% (*w*/*w*) SBO + 2% (*w*/*w*) RJ compared to BPH were significantly different for, respectively, 15 and 18 acids out of the 24 identified acids after a half year’s storage at +4 °C (Table 5).

The bee products used in the study included pollen mixed with honey at a ratio of 1:2 g/g, represented as BPH; pollen mixed with honey at a ratio of 1:2 + 1% (*w*/*w*) SBO, indicated as BPH + 1% (*w*/*w*) SBO); and pollen mixed with honey at a ratio of 1:2 + 1% (*w*/*w*) SBO + 2% RJ, indicated as BPH + 1% (*w*/*w*) SBO + 2% (*w*/*w*) RJ).

C20:4n-6 demonstrated the highest decrease from 1.40% to 0.28% and 0.18% in the samples and BPH + 1% (*w*/*w*) SBO + 2% (*w*/*w*) RJ compared to BPH. The remaining quantity of C20:4n-6 in the latter mixtures amounts to 19.7%. The content of C16:1n-7 increased more than 18.0 times. The content for six FAs did not change significantly in BPH + 1% SBO and BPH + 1% (*w*/*w*) SBO + 2% (*w*/*w*) RJ compared to BPH, and EPA and DHA were not found in the produced mixtures. The data showed that SBO added to honey–pollen mixture reduced the content of ALA. The content of C18:3n-3 in both samples BPH + 1% SBO and BPH + 1% (*w*/*w*) SBO + 2% (*w*/*w*) RJ was nearly equal—25.07% and 25.88%, respectively—and this accounts for 63.0% of BPH present.

#### 3.3.2. Comparative Content of Fatty Acids of Pollen and Pollen Preserved with Honey and Their Mixtures with SBO and RJ during Two Years of Storage

The contents of FAs in the samples BPH + 1% SBO and BPH + 1% (*w*/*w*) SBO + 2% (*w*/*w*) RJ compared to BPH were significantly different for, respectively, 12 and 11 of the 22 identified FAs after two years of storage at +4 °C (Appendix A). The contents of FAs C12:0, C14:10, C16:0, C18:3n-3 and C20:3n-3 were significantly lower in both samples BPH + 1% SBO and BPH + 1% (*w*/*w*) SBO + 2% (*w*/*w*) RJ than in BPH. Fatty acids C16:1n-7, C18:1n-9*cis*, C18:1n-7, C18:2n-6*cis* and C22:0 were found in higher amounts than in BPH.

The amount of C18:3n-3 was lower by 63.0% and 65.0% in the samples BPH + 1% (*w*/*w*) SBO and BPH + 1% (*w*/*w*) SBO + 2% (*w*/*w*) RJ compared to BPH. According to our analysis, adding SBO and RJ to the pollen–honey samples increased the content of FAs C16:1n-7, C18:1n-9*cis* and C18:2n-6*cis* up to five-fold when compared to BPH. The same trend of variation in the contents of the latter FAs was observed when comparing their content in BPH and BPH with an added 1% SBO + 2% RJ after half a year of storage at +4 °C. The differences for all FA contents between samples BPH + 1% SBO and BPH + 1% (*w*/*w*) SBO + 2% (*w*/*w*) RJ were not significant.

Our data revealed a palatable ratio of n-3/n-6 FAs during sample storage for BP and BPH constituted of 3.02 to 3.09, respectively (Table 6). The values of the FAs ratio n-3/n-6 were from 0.79 to 0.91 for BPH + 1% SBO and BPH + 1% SBO + 2% RJ. The data indicate that BP and BPH are suitable products for the human diet; the values of n-3/n-6 in BPH + 1% SBO and BPH + 1% (*w*/*w*) SBO + 2% (*w*/*w*) RJ were lower, those still close to those suitable according to dietology. The content of USFA acids was higher compared to SFA in all tested samples. The content of USFA in BP and BPH compared to SFA was more than two times higher, and in BPH + 1% SBO and BPH + 1% (*w*/*w*) SBO + 2% (*w*/*w*) RJ, it was three times higher. The total sum of n-9 fatty acids was higher than that of n-7 in all samples. The ratio values of n-9/n-7 ranged as follows: 4.34, 4.23, 1.78 and 1.85 in BP, BPH, BPH + 1% SBO and BPH + 1% (*w*/*w*) SBO + 2% (*w*/*w*) RJ after half year, and 4.51, 2.90, 1.56 and 1.60 after two years of storage. The total amount of SFA was mostly the same in all samples after half a year and two years of storage and varied in the range of 24.17–30.62%, as well as 64.22–71.64% for for USFA.

#### 3.3.3. The Relationship between Bee Pollen and Bee Pollen Mixed with Honey during Two Year of Storage

Twenty-two FAs along with eight SFAs and fourteen USFAs were found in BP and BPH after storing them for two years at +4 °C; however, it was lower by seven acids compared to RJ (Appendix A). A strong positive correlation was found between 12 FAs in BP and BPH (*r* = 0.76 and 0.99, respectively); a moderate correlation of 0.43 and 0.59 was estimated between two FAs in the same BP and BPH samples; and a weak correlation (0.27; 0.31 and 0.39) was found for three FAs in BP and BPH samples. The association was negative and moderate for three FAs, with the correlation coefficient (*r*) of −0.47, −0.48 and −0.59. The weakest correlation (*r*) (−0.28 and −0.36) was found for stearic and palmitic FAs. Seventeen out of twenty-two FAs strongly or moderately correlated with BP and BPH, suggesting that the pure pollen has a bigger effect on the content of FAs in BPH than honey.

## 4. Discussion

Royal jelly (RJ) is a secretion produced by the hypopharyngeal and mandibular glands of worker honeybees. Bee queens’ larvae are fed with RJ constantly and therefore develop into reproductive queens; while workers’ larvae are fed only for the first three days [17]. Subsequently, worker bee larvae are fed with worker jelly (WJ), which has a different composition than RJ and is also known as bee queen’s jelly. Significant differences in levels of moisture, protein, 10-hydroxy-2-decenoic acid (10-HDA), fructose (F) and glucose (G) were found between the RJ and WJ samples [52]. RJ is used in health foods and traditional medicines [53].

Research indicates that RJ contains a set of C8, C10 and C12 FAs [54,55]. We identified and quantified caprylic (octanoic) acid (C8:0) in RJ. Medium-chain hydroxy caprylic FAs (C8:0-OH) in the form of 8-hydroxyoctanoic acid are also found in Polish herbal honey [56]. Medium-chain FAs are natural components of coconut oil, palm kernel oil and milk, e.g., goat milk, and characterized by saturated forms of caproic C6:0, caprylic C8:0 and capric C10:0 FAs [57,58].

The lipid content of RJ ranged from 7.0% to 18.0% and was mainly composed of hydroxy FAs with 8–12 carbon atoms, which accounted for 90% of the total lipid content [59]. It has long been considered that the main FA in royal jelly is 10-hydroxy-2-decenoic acid (10-HDA) [21]. The content of 10-HDA in RJ varies depending on the plant source that the bees forage during the production of royal jelly, as well as the bee strain and other factors [60]. However, it is doubtful whether this acid can be used to evaluate the freshness of royal jelly or as a marker with which to assess the quality of royal jelly [61]. According to the requirements of international trade, fresh royal jelly must contain no less than 1.8% of 10-HDA [60].

Fresh RJ is sensitive to light, heat and air. Therefore, special precautions must be taken to preserve the biological properties of RJ during the shelf period [62]. As a result, recommendations are given regarding the storage conditions of RJ and its preparation for longer storage. The requirements are as follows: after collection, RJ should be immediately transferred into a dark and airtight container. If RJ is intended for quick use, it can be stored in refrigerator at 0–5 °C, while for longer storage, RJ should be placed in a freezer at the temperature of −18 °C or below. It is also stated that since there are no criteria for determining the “safety” limits for product activity, storage and shelf-life should be as short as possible. During storage, RJ must be packed in dark, air-tight containers that are tightly closed.

Our study indicated that during the RJ storage for 1.5 years, out of 30 identified FAs, the amount of 13 FAs had been reduced significantly. The remaining content for two FAs was higher than 50.0%. However, for nine FAs, the content reduced from 39.3% to 3.14%. The smallest amount of FA remained for the following acids: C22:5n-3; C16:1n-7*trans*; and C20:3n-3. The residual content of the latter acids was less than 10.0%.

We identified and quantified 8 saturated and 22 unsaturated FAs in fresh RJ. The six long-chain saturated fatty acids are as follows: C14:0; C15:0; C16:0; C18:0; C22:0; and C24:0 (Table 1 and Table 2). The analysis revealed a variety of long-chain USFAs in RJ (Table 2). The α-linolenic acid (ALA; 18:2n-3) and linoleic acid (LA; 18:2n-6*cis*) were found in contents of 4.35% and 5.83%, respectively. The ratio of n-3/n-6 for these long-chain polyunsaturated fatty acids (PUFAs) accounted for 0.75. It is recommended to use food containing PUFAs in the n-6/n-3 ratio, which is 4–5/1 [63]. According to our data, the total content of n-6 PUFAs exceed n-3 by 1.8 times (Table 3). The following USFA from the n-3 family were also identified in RJ: eicosapentaenoic acid (EPA); docosahexaenoic (DHA); eicosatrienoic; and docosapentaenoic. This indicates a proper FAs balance in RJ. The major n-3 FA in the Western diet is ALA [64]. The author confirms that Western diets lack eicosapentaenoic and docosahexaenoic acids (EPA and DHA). According to our previous research, EPA and DHA were determined in bee bread [65].

When storing RJ in the prescribed conditions for 18 months, two FAs had more than 50.0% remaining content, and the other two were close to 50.0%. However, nine FAs were reduced from 39.3% to 3.14%. The smallest amount remained for the following FAs: C22:5n-3; C16:1n-7*trans*; and C20:3n-3. The residual content of these FAs was less than 10.0%.

The buckthorn oil we used in this study contained 94.5-fold higher amounts of linoleic acid (LA; 18:2n-6) than n-3 PUFAs (Table 3). In order to maintain a balanced ratio of n-3/n-6 FA of SBO, which is produced with high content of linoleic acid, it should be mixed with other types of oil that are higher in n-3 PUFAs. The balanced ratio of LA to ALA fatty acids can be improved by using flaxseed oil, which is the richest source of α-linolenic acid (C18:3, omega–3), accounting ˃50.0% of all edible oils. In order to obtain high-quality oil with 1:1 or 1:2 ratios of n-3 and n-6 FAs, selected flax varieties can be sown. [66].

EPA and DHA are nutritionally significant long-chain PUFA produced by the microalgal species [67]. Therefore, microalgal are considered as aquaculture plant feed [68]. These FAs, sometimes called marine omega-3s, are found in seafood (especially fatty fish) and are also used in pharmaceutical products. EPA and DHA are physiologically active acids [69]. Our studies have shown that these acids are also present in some bee products.

Seeds of flax (*Linum usitatissimum*) and camelina (*Camelina sativa*) contain high amounts of ALA, while linoleic acid (C18:2n-6*cis*) was found in its highest concentration in hemp (*Cannabis sativa*) seeds [70]. In modern agriculture, animal feeds have been largely replaced by plant-based feeds to enhance production. As a result, the ratio of FAs required by the human body has changed in many food products. A suitable FA ratio for human health is considered to be n-3/n-6 FA of 1:1 or 1:2, and it can be used as a supplement for a balanced diet [27]. According to other data, it is indicated that an FA ratio of n-6/n-3 from 1:1 to 5:1 is optimal for human health [71]. Three known EFAs—C15:0, α-linolenic (omega-3) and linoleic acid (omega-6)—must be obtained from the human diet because human body does not produce them naturally [72].

Our study demonstrates differences in the concentration of USFA in BP and BPH compared to prepared mixtures. Total n-3 FAs content was found to exceed n-6 content by about three times in BP as well as BPH in the first year and after two years of storage (Table 6). The data show that the level of n-3 FAs was slightly lower in BPH + 1% SBO samples compared to BPH + 1% (*w*/*w*) SBO + 2% (*w*/*w*) RJ samples, where the n-6/n-3 ratio varied in the range 0.79–0.91. The results suggest that by adding the SBO to the mixtures, the content of n-3 FAs can be reduced.

Honeybee products contain different components, and the activity of a honeybee product results from synergic effects of all its components [73]. Research suggests that stingless bees produce honey with antibacterial properties that has no peroxide activity [74]. The antibacterial properties of honey are related to its peroxide and non-peroxide activity. Peroxide-related honey activities were described previously [73]. The most specific components of honey are proteins of honeybee origin. Authentic honeybee proteins found in honey are royal jelly proteins, which include nine members and are designated as MRJP 1–9. The most abundant protein of this family is oligomer MRJP1 in complex with apisimin or its monomer, also known as apalbumin1 or royalactin. MRJP1oligomer protein has also been identified in bee bread [75]. MRJP1 oligomer in complex with apisimin has molecular mass of 280–420 kDa, and MRJP1 monomer has a 55 kDa [8,76].

The monomer of MRJP1, known as royalactin, is reported to be the major factor inducing differentiation in queen bees and honeybees [77]. Several studies have demonstrated the antibacterial properties of royalisins and suggested their uses as potential antimicrobial natural peptides. RJ proteins and peptides have been proposed to be responsible for the immunostimulating properties and antibiotic activity of honey. [78,79]. Previously, we identified the main proteins of plant and the honeybee *Apis mellifera carnica* origin using different mass spectrometry (MS) techniques. The data revealed a range of RJ proteins MRJP1–MRJP9 with a molecular mass of 45.0 kDa to 70.24 kDa in buckwheat honey [80]. The activities of enzymes catalase and glucose oxidase were estimated on the native PAGE in buckwheat honey. Therefore, we can hypothesize that the RJ proteins present in buckwheat honey supplemented its antibacterial properties. The same RJ proteins (MRJP1–MRJP2) were also detected in rape seed (*Brassica napus* L.) honey by our research group [81]. Each RJ protein found in rape seed honey has a different molecular weight than in buckwheat honey. Differentiation among RJ proteins indicates their relationship with the botanical origin of honey [82].

The honey produced by *A. mellifera* contains minor components derived from the nectar collected from plants, including different antimicrobial peptides of bee origin; therefore, honey cannot be considered simply a carbohydrates-rich food [83]. When used for preserving bee products, honey protects the activity of glucose oxidase during storage [13]. The quality of honey is associated with its authenticity. Therefore, one of the RJ proteins, identified as Apalbumin1 and also known as Apa1 or Major Royal Jelly Protein 1 (MRJP1), has been proposed as a marker for honey authenticity and quality [84]. The evaluation of protein diversity and the use of these ingredients as markers to assess honey naturalness is the subject of modern research [84]. Consumers usually prefer mostly monofloral honey and seek information about its quality. Therefore, various studies of honey components and all other bee products can be informative in order to evaluate their quality. Biomarkers such as EPA and DHA are proposed to assess the authenticity and biological value of bee products for functional food.

## 5. Conclusions

The study shows that the highest FAs diversity was found in RJ compared to BP and BP prepared for storage with honey, as well as SBO and RJ. Essential fatty acids EPA and DHA were found in RJ as well as in BP and BP mixed with honey. These FAs were not found in the samples prepared with SBO, even in the sample supplemented with RJ. The SBO used in this study was found to be high in linoleic acid, resulting in an increased n-6 fatty acids ratio in the prepared samples.

During the RJ storage for 1.5 years at −20 °C, out of 30 identified FAs, the content of 13 FAs had reduced significantly. Docosadienoic FA decreased the most in sample BPH compared to BP and arachidonic acid in sample BPH + 1% SBO compared to BPH + 1% (*w*/*w*) SBO + 2% (*w*/*w*) RJ when these samples were stored under the intended conditions. The amount of these acids decreased significantly after half a year of storage. Differences in total FAs content between samples BP and BPH as well as BPH + 1% SBO and BPH + 1% (*w*/*w*) SBO + 2% (*w*/*w*) RJ were insignificant.

Bee-collected pollen had the greatest influence on the number of FAs in its mixture with honey.

## Figures and Tables

**Figure 1 foods-12-03164-f001:**
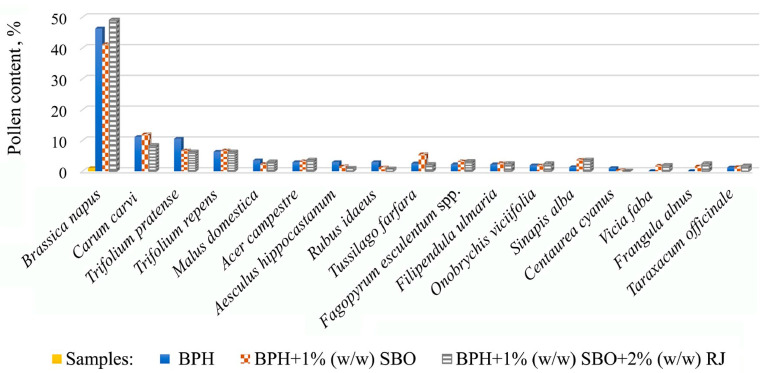
Pollen content in samples.

**Figure 2 foods-12-03164-f002:**
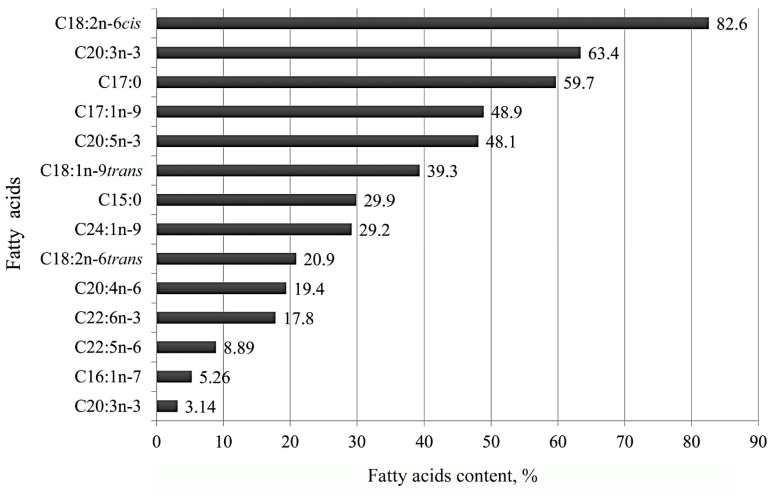
Residual contents of fatty acids in royal jelly after 18 months of storage at −20 °C.

**Table 1 foods-12-03164-t001:** SFA composition (% of total FAs), basic statistics and fatty acids retention time (RT) of sea buckthorn oil (SBO) and royal jelly (RJ) used as additives to bee pollen mixed with honey.

Fatty Acids	SFA Formula	SBO	RJ	*p* < 0.05 Value	CV, % SBO/RJ	RT Min
Caprylic	C8:0	0	1.54 ± 0.07	0.07	7.57	4.31
Myristic	C14:0	0.18 ± 0.01	0.75 ±0.03	0.57	6.30/6.55	22.55
Pentadecanoic	C15:0	0.04 ± 0.00	0.32 ± 0.02	0.29	15.75/9.45	24.30
Palmitic	C16:0	12.12 ± 0.10	15.36 ± 0.25	0.76	1.40/2.76	26.03
Margaric (Heptadecanoic)	C17:0	0.04 ± 0.01	0	-	13.32	27.72
Stearic	C18:0	3.37 ± 0.07	20.68 ± 0.31	1.65	3.64/2.62	29.37
Arachidic	C20:0	0.25 ± 0.02	0.11 ± 0.01	0.10	11.7/15.75	32.56
Behenic	C22:0	0.52 ± 0.52	0.44 ± 0.02	0.16	9.15/7.00	35.85
Lignoceric	C24:0	0.15 ± 0.01	0.62 ±0.05	0.22	7.87/12.90	39.43

**Table 2 foods-12-03164-t002:** USFA composition (% of total FAs), basic statistics and fatty acids retention time (RT) of sea buckthorn oil (SBO) and royal jelly (RJ) used as additives to bee pollen mixed with honey (*p* ≤ 0.05).

Fatty Acids	USFA Formula	SBO	RJ	*p* < 0.05 Value	CV, % SBO/RJ	RT Min
Palmitelaidic	C16:1n-7*trans*	0	0.19 ± 0.01	-	-/9.12	26.95
Palmitoleic	C16:1n-7	9.11 ± 0.12	1.13 ± 0.06	0.25	2.24/9.37	27.37
Hexadecenoic	C16:1n-9	0	0.33 ± 0.02	-	-/9.37	27.16
Margaric (Heptadecanoic)	C17:1n-9	0.32 ± 0.01	0.16 ± 0.01	0.14	6.44/14.74	28.88
Vaccenic	C18:1n-7	2.20 ± 0.02	0.57 ± 0.02	0.12	11.17/4.64	30.65
Elaidic	C18:1n-9*trans*	0.02 ± 0.00	0.58 ± 0.02	0.08	2.94/0.86	30.16
Oleic	C18:1n-9*cis*	13.03 ± 0.15	21.30 ± 0.11	0.19	1.93/0.86	30.51
α-Linolenic (ALA)	C18:3n-3	0.43 ± 0.02	4.35 ± 0.09	0.46	6.66/3.73	34.49
Linoleic	C18:2n-6*cis*	57.16 ± 0.09	5.83 ± 0.09	0.44	0.27/2.72	32.19
Linolelaidic	C18:2n-6*trans*	0.41 ± 0.03	0.43 ± 0.02	0.14	11.2/6.98	34.18
γ-Linoleic	C18:3n-6	0	0.95 ± 0.03	-	-/5.82	33.49
Eicosenoic	C20:1n-9	0.12 ± 0.00	0.29 ± 0.02	0.07	3.43/10.07	33.74
Eicosadienoic	C20:2n-6	0.07 ± 0.07	0.10 ± 0.01	0.07	8.66/22.35	35.51
Eicosatrienoic	C20:3n-3	0	0.93 ± 0.01	-	-/2.15	37.66
Eicosapentaenoic (EPA)	C20:5n-3	0.18 ± 0.02	0.27 ± 0.01	0.09	16.67/3.70	40.58
Eicosatrienoic	C20:3n-6	0	1.06 ± 0.05	-	-/7.37	36.92
Arachidonic	C20:4n-6	0.03 ± 0.00	3.40 ± 0.11	0.45	21.65/5.42	38.07
Docosadienoic	C22:2n-6	0	0.25 ± 0.01	-	-/6.03	39.10
Docosatetraenoic	C22:4n-6	0	0.52 ± 0.02	-	-/7.23	42.39
Docosopentaenoic	C22:5n-3	0	1.39 ± 0.05	-	-/6.85	45.46
Docosahexaenoic acid (DHA)	C22:6n-3	0	0.69 ± 0.01	-	-/3.0	46.90
Nervonic	C24:1n-9	0.10 ±.00	0.17 ± 0.00	0.01	5.59/3.46	40.58

Note: USF—unsaturated fatty acids: n-3; n-6; n-7; n-9; n-3: C18:3n3; C20:3n3; C20:5n3; C22:5n3; C22:6n3; n-6: C18:2n6; C18:2n6*trans*; C18:3n6; C20:2n6; C20:3n6; C20:4n6; C22:2n6; C22:4n6; n-7: C16:1n7; C16:1n7*trans*; C18:1n7; n-9: C16:1n9; C17:1n9; C18:1n9*trans*; C18:1n9*cis*; C20:1n9; C24:1n9.

**Table 3 foods-12-03164-t003:** The total sum and range of saturated and unsaturated fatty acids (% of total FA) in the sea buckthorn oil (SBO) and royal jelly (RJ).

Fatty Acids	SBO			RJ		
	Total Sum	Min	Max	Total Sum	Min	Max
n-3	0.61	0.15	0.45	6.94	0.26	4.48
n-6	57.67	0.02	57.34	12.54	0.09	6.0
n-7	11.31	2.17	9.26	1.89	0.18	1.21
n-9	13.59	0.02	13.32	22.83	0.17	21.50
Saturated FA	16.63	0.10	21.04	39.82	0.10	21.04
Unsaturated FA	83.18	0.02	57.34	44.20	0.09	21.50
n-3/n-6	0.01			0.90		
n-6/n-3	94.5			1.81		
n-7/n-9	0.83			0.08		
n-9/n-7	1.20			12.10		
SFA/UFA	0.20			0.91		
USFA/SFA	5.00			1.11		

**Table 4 foods-12-03164-t004:** The influence of honey addition to the mixture with pollen on the fatty acids content (% of total FA) after 6 months of storage.

Fatty Acids	Formula	BP	BPH	*p* < 0.05 Value
Lauric	C12:0	0.53	0.23	0.13
Myristic	C14:0	1.54	1.50	0.30
Pentadecanoic	C15:0	0.88	0.96	0.08
Palmitic	C16:0	21.91	24.12	2.46
Palmitoleic	C16:1n-7*trans*	0.16	0.19	0.08
Stearic	C18:0	1.73	2.56	0.11
Oleic	C18:1n-9*cis*	5.87	6.55	0.21
Vaccenic	C18:1n-7	1.32	1.49	0.15
Linolelaidic	C18:2n-6*trans*	2.10	1.50	0.28
Linoleic	C18:2n-6 *cis*	10.16	9.76	0.19
γ-Linoleic	C18:3n-6	0.32	0.25	0.17
α-Linolenic	C18:3n-3	43.0	40.24	0.99
Arachidic	C20:0	0.70	0.56	0.12
Eicosenoic	C20:1n-9	0.16	0.17	0.03
Eicosadienoic	C20:2n-6	0.17	0.19	0.13
Eicosatrienoic	C20:3n-3	0.90	0.51	0.12
Eicosapentaenoic (EPA)	C20:5n-3	0.34	0.33	0.23
Arachidonic	C20:4n-6	0.74	1.40	0.08
Behenic	C22:0	0.34	0.33	0.23
Docosadienoic	C22:2n-6	0.67	0.32	0.24
Docosatetraenoic	C22:4n-6	0.34	0.39	0.23
Docosahexaenoic (DHA)	C22:6n-3	0.19	0.93	0.11
Lignoceric	C24:0	0.31	0.36	0.17
Nervonic	C24:1n-9	0.39	0.39	0.19

Note: SFA—C12:0; C14:0; C15:0; C16:0; C18:0; C20:0; C22:0; C24:0; USF: n-3; n-6; n-7; n-9; C24:1. n-3: C18:3n3; C20:3n3; C20:5n3; C22:6n3; n-6: C18:2n6*trans*; C18:2n6*cis*; C18:3n6; C20:2n6; C20:4n6; C22:2n6; C22:4n6; n-7: C16:1n7*trans*; C18:1n7 and n-9: C18:1n9; C20:1n9; BP—bee pollen; BPH—pollen mixed with honey at a ratio of 1:2.

**Table 5 foods-12-03164-t005:** Comparison of fatty acids content (% of total FA) of pollen preserved with honey and their mixtures with sea buckthorn oil (SBO) and royal jelly (RJ) after half a year storage at +4 °C.

Fatty Acids	Formula	BPH	BPH + 1% SBO	BPH + 1% SBO + 2% RJ	*p* < 0.05 Value
Lauric	C12:0	0.23	0.13	0.13	0.04
Myristic	C14:0	1.50	0.94	1.03	0.17
Pentadecanoic	C15:0	0.96	0.38	0.42	0.06
Palmitic	C16:0	24.12	18.67	18.64	0.96
Palmitoleic	C16:1n-7	0.19	3.49	3.33	0.26
Stearic	C18:0	2.56	2.49	2.69	0.14
Oleic	C18:1n-9*cis*	6.55	8.64	8.89	0.29
Vaccenic	C18:1n-7	1.49	1.59	1.70	0.14
Linolelaidic	C18:2n-6*trans*	1.50	1.10	1.12	0.25
Linoleic	C18:2n-6	9.76	29.91	29.34	0.16
γ-Linoleic	C18:3n-6	0.25	0.17	0.14	0.03
α-Linolenic	C18:3n-3	40.24	25.07	25.28	0.40
Arachidic	C20:0	0.56	0.52	0.54	0.09
Eicosenoic	C20:1n-9	0.17	0.19	0.15	0.54
Eicosadienoic	C20:2n-6	0.19	0.22	0.22	0.09
Eicosatrienoic	C20:3n-3	0.51	0.29	0.26	0.09
Arachidonic	C20:4n-6	1.40	0.28	0.18	0.08
Eicosapentaenoic (EPA)	C20:5n-3	0.33	0	0	
Behenic	C22:0	0.34	0.44	0.46	0.11
Docosadienoic	C22:2n-6	0.32	0.26	0.16	0.11
Docosatetraenoic	C22:4n6	0.39	0.20	0.25	0.11
Docosahexaenoic (DHA)	C22:6n-3	0.93	0	0	0
Lignocerin	C24:0	0.36	0.25	0.26	0.08
Nervonic	C24:1n-9	0.40	0.23	0.26	0.07

Note: SFA—C12:0; C14:0; C15:0; C16:0; C18:0; C20:0; C22:0; C24:0; USF: n-3; n-6; n-7; n-9; C24:1.

**Table 6 foods-12-03164-t006:** The total sum and ratio of unsaturated (USFA) and saturated (SFA) fatty acids (% of total FAs) in bee pollen mixed with honey and their mixtures with sea buckthorn oil (SBO) and royal jelly (RJ) during storage for 6 months and after two years (*p* ≤ 0.05), where average FAs content after 6 months of storage is indicated with ^1^ and FAs content after two years of storage is indicated with ^2^.

Groups of Fatty Acids	BP ^1^	BP ^2^	BPH ^1^	BPH ^2^	BPH + 1% SBO ^1^	BPH + 1% SBO ^2^	BPH + 1% SBO + 2% RJ ^1^	BPH + 1% SBO + 2% RJ ^2^
n-3	44.43	42.0	42.01	42.72	25.36	26.76	25.54	27.69
n-6	14.43	12.99	13.81	13.79	32.14	31.12	31.41	30.28
n-7	1.48	1.74	1.68	2.36	5.08	5.53	5.03	5.50
n-9	6.42	7.85	7.11	6.84	9.06	8.62	9.30	8.77
Saturated FA	27.94	29.12	30.62	29.59	23.82	24.21	24.17	24.41
Unsaturated FA	66.76	64.57	64.22	65.71	71.64	72.03	71.28	24.41
n-3/n-6	3.08	3.02	3.04	3.09	0.79	0.86	0.81	0.91
n-7/n-9	0.23	0.22	0.24	0.35	0.56	0.64	0.54	0.63
n-9/n-7	4.34	4.51	4.23	2.90	1.78	1.56	1.85	1.60
SFA/UFA	0.42	0.45	0.48	0.45	0.33	0.34	0.34	0.34
USF/SFA	2.39	2.22	2.10	2.22	3.01	2.98	2.95	2.96

Note: the bee products used in the study included pollen mixed with honey at a ratio of 1:2 g/g, represented as BPH; pollen mixed with honey at ratio 1:2 + 1% (*w*/*w*) SBO, indicated as BPH + 1% (*w*/*w*) SBO; and pollen mixed with honey at a ratio 1:2 + 1% (*w*/*w*) SBO + 2% RJ, indicated as BPH + 1% (*w*/*w*) SBO + 2% (*w*/*w*) RJ.

## Data Availability

Datasets analysed during the current study are available from the authors on reasonable request.

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
