# Peer review of "The Composition of Fatty Acids in Bee Pollen, Royal Jelly, Buckthorn Oil and Their Mixtures with Pollen Preserved for Storage"

_foods, 2023, doi:10.3390/foods12173164_

Round 1

Reviewer 1 Report

Your manuscript “The Composition of Fatty Acids in Bee Pollen, Royal Jelly, Buckthorn oil and Their Mixtures with Pollen Preserved for Storage” provides interesting information on the composition of FAs in different products derived from the beehive and other additives (alone and in mixtures), as well as its changes during storage at different periods and temperatures.

During the review of the manuscript, some aspects have been detected that could be considered to improve the document, which are described below, and indicated in the PDF file.

Suggestions

L70: Decanedioic acid and 2-decenoic acid were…

L72: Remove parentheses “ )

L91: enlongase enzyme? Should it be elongase? Check, please

L111: Use FAs (abbreviation) or fatty acids, not both. Check this same in L174, L205

L142: samle? Or sample? Check, please

L148: Indicate the highlighted information in brackets: Sea buckthorn oil was purchased at a pharmacy (Sea buckthorn oil producer SALVENA S.C., Poland) and stored at +4 °C.

L170: Remove the dash before the value of the temperature of the injector, detector and carrier gas flow; they could be read as negative values.

L172: Delete marked text; FAME mix characteristics have been indicated in “Chemicals”

L190: In Figure 1, “Samples:” is part of the plotted series? If not, eliminate it from the plot. In any case, this word would be the title of the X-axis, but it should not be indicated in the current format of Figure 1.

L198: Same as L170

L216: Notes on the information in the table; it is recommended to indicate them at the bottom of the table.

L219: Check the numbering of the subsections from 3.2; would this be 3.2.1?

In Tables 1, 4 and 5, the authors mean Lignoceric? Check, please

L224-225: Same as L216

L271: In Figure 2, insert the X-axis title in the graph.

L290: Same as 216

L315-317; L362-366: Too much information in a Table title. Proceed as in L216

L363: w/w, as line 15 and L142

-Finally, in Supplementary material PDF, also review the title of Tables S1 and S2, as well as the name of lignoceric acid:

Table S1. Comparative average of fatty acids (FA) content (% of total FA) of pollen preserved with honey and their mixtures with sea buckthorn oil (SBO) and royal jelly (RJ) in a two-year period of storage at +4 °C (p ≤ 0.05). The bee products used in the study included pollen mixed with honey in ratio 1:2 g/g represented as BPH, pollen mixed with honey in ratio 1:2+1 % (w/w) SBO indicated as BPH+1 % (w/w) SBO), pollen mixed with honey in ratio 1:2+1 % (w/w) SBO+2 % RJ indicated as BPH+1% (w/w) SBO+2 % (w/w) RJ)

Table S2. Relationship of fatty acids (% of total FA) between bee pollen and honey mixture with pollen during a two-year period of storage at +4 °C (p ≤ 0.05), where CV indicates coefficient of variation and SE – standard error. The bee products used in the study included pollen mixed with honey in ratio 1:2 g/g represented as BPH

Reviewer 2 Report

As a reviewer, I find the paper entitled "The Composition of Fatty Acids in Bee Pollen, Royal Jelly, Buckthorn oil and Their Mixtures with Pollen Preserved for Storage" to be highly valuable and informative. The study significantly contributes to our understanding of the fatty acid (FA) composition of bee products, particularly under storage conditions. However, there are some areas in the manuscript that require improvements, and I have provided detailed comments and suggestions in the attached file.

I hope these comments and suggestions will help the authors enhance the manuscript and contribute to the overall quality and impact of the paper.

The manuscript requires extensive editing and improvement related to the English language. There are issues with grammar, sentence structure, and overall clarity that need to be addressed. Additionally, some sections may require rephrasing or reorganizing to enhance the flow and readability of the content.

Round 2

Reviewer 2 Report

I am happy to see that the current version of the manuscript has been improved a lot by the authors an I can suggest the paper for publication.